# Maximizing Collagen Yield from Underutilized Lumpfish (*Cyclopterus lumpus*) Skins by Optimizing Pre-Cleaning and Extraction Methods

**DOI:** 10.3390/md22120525

**Published:** 2024-11-22

**Authors:** Judith Maria Scheja, Jens Jakob Sigurðarson, Halldór Gunnar Ólafsson, Hjörleifur Einarsson

**Affiliations:** 1BioPol ehf, 545 Skagaströnd, Iceland; judith@biopol.is (J.M.S.); jens@biopol.is (J.J.S.); halldor@biopol.is (H.G.Ó.); 2Faculty of Natural Resource Sciences, University of Akureyri, 600 Akureyri, Iceland

**Keywords:** side streams, by-products, rest raw material, waste, gelatin, scale-up

## Abstract

Female lumpfish (*Cyclopterus lumpus*) are a primary target of commercial fishery for their roe, a substitute for caviar. The remaining carcasses are underutilized rest raw material. The pre-treatment and acid extraction conditions of collagen from lumpfish skins were optimized. Full factorial design was used to optimize the alkali pre-treatment conditions with NaOH. The optimal conditions were X_1_ = 0.1 M (NaOH concentration), X_2_ = 6 h (NaOH treatment time), X_3_ = 4 °C (treatment temperature) and X_4_ = 1:5 (*w*/*v*, solid to liquid ratio). Optimized conditions for collagen extraction with acetic acid were investigated using a Box–Behnken design. The result suggested a concentration of 0.9 M acetic acid, treatment temperature of 21 °C, a treatment time of 36 h in a total of 2 volumes of acid. Combined optimized effects resulted in a collagen yield of 45% (DW/DW) with high purity (>90%) and a high hydroxyproline content (7.9%). A scale-up experiment (starting with 45 kg) showed that the yield was somewhat lower (18–25%). Enzyme hydrolysis of skin after acetic acid extraction added another 23% (DW/DW) to the yield and enzyme hydrolysis of precleaned skins resulted in 60% (DW/DW).

## 1. Introduction

The fishery industry is an important economic sector with a global harvest of 178 million tons in 2020, where approximately 90 million tons are wild capture [1]. Production and processing in the fish industry results in large amounts of fish side streams and wasted by-products [2]. These underutilized by-products are potential sources of valuable proteins, oils, vitamins, and minerals [3].

The lumpfish (*Cyclopterus lumpus*) is a marine cold-water fish found in the North Atlantic [4]. Female lumpfish are caught for their roe, a substitute for caviar. In 2023, the total catch of lumpfish in Iceland was 4000 tons [5], generating around 2400 tons of underutilized side-streams (corresponding to 60% of the catch). Fish skin in general is a valuable side stream, as it contains 10–15% proteins. Collagen is a structural protein used in a wide range of industries: it can be used as a food supplement, incorporated in cosmetics, and act as a biomaterial in pharmaceutics [6]. The collagen content of lumpfish skin from aquaculture was found to be around 20%, compared to 13% in Atlantic cod skin and 25% in Atlantic salmon skin (on dry weight basis) [7]. Gelatin (heat-extracted collagen) was found to be 14% dry weight (1.4% wet weight) in wild caught lumpfish [8]. In a recent study on wild caught lumpfish from Iceland, collagen content of 37% was reported [9].

The abundance of lumpfish side streams generated every year, the relatively high amount of collagen found in lumpfish skin, and the interest in non-mammalian collagen make lumpfish skin an attractive source of collagen. Collagen is currently mainly obtained from animals like cows, pigs and chickens [6]. Marine collagen has similar properties to mammalian collagen [10]. The collagen molecule consists of three left-handed α-helices that form a right-handed superhelix [11]. Each polypeptide chain is made up of a repeated sequence (Gly-X-Y-), most commonly with proline in X and 4-hydroxyproline in Y-position. Hydroxyproline is found in collagen and in small amounts in elastin but is nearly absent in all other proteins. Hence, it is used as a marker for the identification and quantification of collagen [11,12,13].

The extraction of collagen from fish skins consists of three main steps: (I) preparation of the skin raw material (washing, size reduction, removal of non-collagen proteins in alkaline solutions and sometimes demineralization), (II) extraction of the collagen, and (III) recovery and purification, e.g., precipitation of the collagen with NaCl, dialysis, and/or filtration before drying.

The results (i.e., yield of collagen) vary greatly depending on, e.g., species, tissue, and extraction methodology. A recent review by Farooq et al. [14] shows this variation clearly, reporting collagen yield values from less than 1% in shrimp and as high as 82% in Gulf corvina (*Cynoscion othonopterus*) skin. Laasri et al. [15] reported similar yields (13.8–67.3%) of collagen from fish skins. Lumpfish skin is considered a potential source of marine collagen; however, reported collagen content is rather low (19.5% [7], 14% [8], 14.3% [16] and 40% [9]).

The varying results also reflect the different methodology used. Most common collagen extraction methods are acid and enzymatic hydrolysis. Acid Soluble Collagen (ASC) is produced using various acids and Pepsin Soluble Collagen (PSC) with enzymes [12,17,18]. Organic acids (acetic acid, citric acid, and lactic acid) and inorganic acids (hydrochloric acid, formic acid, sulfuric acid, and tartaric acid) can be used [17,19]. Analyses show that organic acids are more efficient than inorganic ones, preferably acetic acid in a concentration of 0.5 M [12,20,21,22,23,24]. Temperatures between 4 °C and 20 °C have been suggested for acid extraction [24], but time and solid-to-liquid radio also have an effect [12].

It was hypothesized that the collagen extraction yield could be improved, and thus the aim of this study was to develop a protocol to maximize the collagen yield from lumpfish skins and thereby enhance the value of by-products from the lumpfish fishing industry.

## 2. Results and Discussions

### 2.1. Body and Proximate Composition Analysis

The average body composition of female lumpfish was determined to consist of the following: roe 40 ± 4%, skin 21 ± 3%, fillets 15 ± 4%, backbone 11 ± 1% and head 14 ± 3%. The lumpfish skin contained 87.9% water, 9% protein, 0.3% fat and 1% ash. In a study by Sato et al. [25], lumpfish skin was reported to contain approximately 86% water and 10% protein, providing a useful baseline for assessing the composition and quality of collagen extraction processes. For juvenile lumpfish from aquaculture, it was found that the skin had a protein content of 6.3% and contained 0.9% fat [20].

### 2.2. Protein and Hydroxyproline Removal by Alkaline Pre-Treatment

Prior to the collagen extraction, the non-collagen proteins were removed through an alkaline pre-treatment. This pre-treatment is commonly performed using NaOH [7,20] in concentrations between 0.1 and 0.5 M NaOH. To achieve the highest extraction yield in addition to the purest product, the optimal conditions of this pre-treatment step were investigated. The success of the pre-treatment was evaluated based on the amount of total protein removed (higher amounts suggesting a more complete pre-treatment) and by the amount of hydroxyproline removed (indicating loss of collagen).

The total protein content removed from 30 g lumpfish skin samples by NaOH treatment is shown in Figure 1a. The higher temperature (21 °C) gave a somewhat greatest removal than the lower (4 °C) one. The higher sample-to-liquid ratio (1:5) also gave a somewhat greater removal than the 1:10 ratio.

An ANOVA found that the factors ratio and temperature had a significant effect on the amount of protein removed during pre-treatment (data shown in Appendix A). The greatest amount of protein was removed in a ratio of 1:10, time 24 h, and temperature 21 °C.

The collagen content is often determined from the total amount of hydroxyproline by using a specific conversion factor based on the relative hydroxyproline content of collagen [26]. The hydroxyproline content measured in the untreated lumpfish skin was used to determine the initial collagen content in the skins and the removed collagen content in the filtrate. Hydroxyproline content in untreated lumpfish skin was found to be 4.4 ± 0.9%, corresponding to 55% collagen (conversion factor 12.6) DW basis.

The removal of collagen from the skin chunks during the pre-treatment, expressed as hydroxyproline content, was measured in the filtrate, and results are shown in Figure 1b. A higher temperature (21 °C) showed more hydroxyproline removal (4.8–71 mg) than at a lower temperature (4 °C) (2.8–7 mg). A stronger NaOH solution (0.5 M) showed greater hydroxyproline removal than lower concentrations (0.1 M). Thus, pre-cleaning at lower temperature and lower NaOH concentration will increase the collagen yield.

An ANOVA found that there was significant interaction between concentration and temperature, between concentration and duration, and between temperature and duration, with regard to the amount of hydroxyproline removed during pre-treatment (*p* < 0.001). The factors temperature (21 °C), time (24 h) and concentration (0.5 M) removed the most hydroxyproline from the lumpfish skins, and 4 °C and 6 h showed the least removal and thus are preferred. Results of the ANOVA suggested that the main factors concentration, temperature and duration have a strong effect on the hydroxyproline content (*p* < 0.001). The factor ratio also showed to have a significant effect on the hydroxyproline content (*p* < 0.05). ANOVA results are shown in Appendix A.

The present study aimed at optimizing the pre-treatment with NaOH of lumpfish skins for the extraction of acid-soluble collagen. The four factors temperature, concentration, duration and ratio of NaOH pre-treatment conditions were tested. The results show that the highest content of both hydroxyproline and non-collagen proteins were removed in the following conditions: temperature 21 °C, duration 24 h, concentration 0.5 M and 10 volumes (*w*/*v*) of NaOH. It has been reported by Sato et al. [25] that the removal of non-collagen proteins from carp muscle with 0.01 and 0.05 M NaOH was not successful. A concentration of 0.5 M and 1.0 M removed a significant amount of collagen. In their study, the best results were obtained by pre-treatment with 0.1 M NaOH, most collagen remained, but non-collagen proteins were removed effectively [25]. The effects of temperature, concentration, duration, and ratio of NaOH on the pre-treatment of pollock skin was studied by Zhou et al. [27], and the authors suggested a pre-treatment temperature lower than 10 °C for cold-water fish species. Therefore, the treatment with 1:5 (*w*/*v*) 0.1 M NaOH at 4 °C for 6 h is sufficient to prepare the skins for acid treatment.

### 2.3. Acid Extraction of Collagen

The results of different extraction times, acid concentrations and sample-to-solution ratios at 21 °C on collagen and hydroxyproline yield (%) are shown in Figure 2. The collagen yield (dry weight) of the samples varied between 1% (at 24 h extraction, 0.9 M acid, and solid-to-liquid (*w*/*v*) ratio 1:8) and 39% (at 36 h extraction, 0.5 M acid, and solid-to-liquid (*w*/*v*) ratio 1:2. Hydroxyproline yield varied between 2.3% dry weight (at 24 h extraction, 0.9 M acid and solid-to-liquid (*w*/*v*) ratio 1:8) and 8.6% (at 12 h extraction, 0.9 M acid, and solid-to-liquid (*w*/*v*) ratio 1:5). The samples extracted at 4 °C resulted in low collagen yields between 0.07% (12 h, 0.1 M and solid-to-liquid (*w*/*v*) ratio 1:5) and 0.8% (36 h, 0.5 M and solid-to-liquid (*w*/*v*) ratio 1:8). The hydroxyproline yields were between 0.17% (12 h, 0.9 M and solid-to-liquid (*w*/*v*) ratio 1:5) and 8.8% (24 h, 0.5 M and solid-to-liquid (*w*/*v*) ratio 1:5).

After screening the samples, further analysis was performed on the samples extracted at 21 °C, as these showed more promising results (on average, higher collagen yield and hydroxyproline content).

#### SDS-PAGE of Collagen Extracts

SDS-PAGE was performed to separate the collagen fibers and identify the collagen type and the purity of the sample. Collagen α- and β-chains are generally used for the identification of the collagen type, with α1 and α2- chains having a molecular weight of around 120–150 kDa and β- chains between 200 and 250 kDa. Collagen type I consists of two α1- and one α2- chains [12,22]. Figure 3 shows the SDS-PAGE patterns of collagen samples extracted at 21 °C from lumpfish skins and indicates the presence of β and α-chains in the samples. Similar patterns for collagen from fish skins were found in [22,28] and for lumpfish collagen by Vate et al [29].

### 2.4. Verification of Found Optimal Alkali Pre-Treatment and Acid Extraction Parameters

The optimal pre-treatment and extraction conditions determined above were tested together on a larger scale (1 kg) of lumpfish skin.

In total, five fractions were obtained after the salting-out process of the collagen (Table 1). The highest purity in hydroxyproline was found in fraction 4 (7.20%), and the lowest in fraction 1 (6.28%). The fractions 1–4 showed a low dry weight (0.03–2.02 g). Fraction 5 showed the highest dry weight (49.2 g or 44.7%). Martinsdottir [8] reported 7.8% hydroxyproline in wild caught (Iceland coast) lumpfish, Osborne [16] reported a slightly lower content (5.7%) and Dave et al. [7] reported a much lower hydroxyproline content (2.4%). The high content of hydroxyproline in the present study is comparable to the content found in cod skins (7.29–7.66%) [30].

Due to the high dry weight (49.2 g) and low hydroxyproline of 6.69%, fraction 5 was subjected to further analyses and purification. The additional purification step of fraction 5 resulted in an increased hydroxyproline content (average 7.92% ± 0.80%) and showed that longer dialysis time is necessary to obtain a higher purity of collagen. It was assumed that the collagen obtained was almost 100% pure. This hydroxyproline % gives a conversion factor of 12.6, which is higher than the 11.6 used by Sasidharan et al. [9] working with lumpfish caught in Iceland.

As shown in Figure 4, the SDS-PAGE patterns of α-chains and β-chains of the five fractions are like that in Figure 3, as well as like those of Vate et al. [29], who studied the structural and functional properties of lumpfish collagen.

### 2.5. Scale-Up Extraction

The results of the scale up experiment are shown in Table 2.

The DW of the starting material was 4862 g. During the NaOH pre-treatment approximately 660 g (14%) was washed out. Collagen was calculated to be 3118 g in the starting material or 64% of DW. This is a little higher than what was found (53%) in the 1 kg experiment. Like the DW yield, the collagen yield also decreases through the pre-treatment and extraction process, but the purity increases. The extraction liquid (AcOH-solution) has approximately 1.3% DW, but during filtration, the DW content increases to 6.6%. The collagen in the extraction liquid was estimated at 1258 g and 872 g in the retentate. The DW of the extraction liquid was 31% of the original DW, and in some studies, this is reported as a collagen yield. The collagen in the extraction liquid was 26% of the initial DW and 40% of the initial collagen. The collagen yield after filtration was found to be 18% of the original DW and 25% of the original collagen. This yield is lower than what was found in the small scale (39%) and 1 kg experiment (44.7%). The table also shows that some 1145 g (37%) of initial collagen is still present in the skin chunks.

Working on a small pilot scale as in this case causes some losses during the process. During the filtration, 240 g DW (~30%) was lost, even with thorough cleaning of filters and tubes. This type of loss should be less prominent when working with still larger volumes. Additionally, this type of loss is almost neglectable in experiments using salting out and dialysis as the collagen recovery process. Thus, those methods are expected to give higher yields, but they are not practical when working with larger volumes.

### 2.6. Enzyme and Acid Extraction of Collagen

As shown above, the skin chunks still contain considerable amounts of collagen after acetic acid extraction. Thus, an experiment using enzymes was carried out to see if recovery could be improved, e.g., by enzymatic hydrolysis. The experiment both studied the effect of acid extraction combined with enzyme digestion and enzyme hydrolysis by itself without acid extraction. The results are shown in Figure 5.

The DW of the lumpfish skin used for this experiment was 15.4 g. The HYP content was 5.93%, and collagen was 75% of the DW. The pre-treated skin had 86% of the initial DW and 70% collagen yield. This material was then subjected to acetic acid extraction and enzyme digestion. The acid soluble collagen (ASC) yield was 51%, and the purity was 90%. This fraction was purified further, resulting in 18% collagen yield with 95% purity. The remaining skin still contained some 19% collagen, and those skin chunks were enzyme-hydrolyzed, yielding an additional 23% of the collagen but with lower purity (84%). This procedure (acetic acid extraction followed by enzyme hydrolysis) yielded, in total, 41% collagen of the initial DW. The yield from skin samples subjected only to enzyme hydrolysis resulted in 60% collagen yield with a purity of 82%.

The results indicate that enzyme hydrolysis of pre-treated lumpfish skin gives higher collagen yield (60% DW/DW) than the combined acetic acid extraction followed by enzyme hydrolysis (41% DW/DW). It should, though, be noted that not all the collagen was recovered in the purification of the AcOH solution. The AcOH extraction gave purer collagen than the extraction by enzyme hydrolysis, especially when AcOH extraction was followed by enzyme hydrolysis. Both approaches left around 2–3% of the initial collagen in the skin chunks. Although the combined effect (acetic acid and enzyme hydrolysis) gave lower recovery than the enzyme hydrolysis alone, the ASC preserves much of the initial structure whereas the enzyme soluble collagen (ESC) is hydrolyzed into smaller units and have lost most of the properties of natural collagen.

In our work, we have found that the initial collagen content of lumpfish skin ranges from 50–75% of skin DW. Our results (41% and 60%) are close to the lower limit, and some increase in yield can still be expected, especially regarding the purification step after the acetic acid extraction.

It was hypothesized that the extraction yield of collagen from lumpfish skin could be increased from the reported values of between 14% and 40%. The protocol developed in this work showed that up to 60% collagen could be extracted.

## 3. Materials and Methods

### 3.1. Preparation of Lumpfish Skins

Frozen lumpfish were fished off the north coast of Iceland and obtained through the Icelandic fishing companies Drangur ehf and Brim hf, Iceland (catch season 2023) and stored at −20 °C until further use. The lumpfish was thawed and manually skinned, and the clean skins were mechanically shredded into small pieces (2 × 2 cm^2^) with an industrial blender.

### 3.2. Alkaline Pre-Treatment

Full factorial design was used to identify the optimal conditions for the pre-treatment of lumpfish skins with NaOH. The independent variables time, concentration, temperature, and ratio (defined as the amount of fish skin divided by the amount of solution) were evaluated at two levels. Experimental range and values of independent variables are shown in Table 3.

Thawed and shredded lumpfish skins (30 g) were soaked in NaOH solution with different pre-treatment times, concentrations, temperatures and ratios, as summarized in Table 3. A total of 16 experiments were performed in duplicates. After the alkali pre-treatment the skins were removed, both pre-treatment solutions and pre-treated fish skins were stored at −20 °C for further analysis. To determine the effects of independent variables on the extraction of collagen the following parameters were examined: total amount of removed protein (g) and content of removed hydroxyproline (mg) in the pre-treatment solutions.

### 3.3. Acid Extraction

Box–Behnken design was used to optimize the acid extraction conditions. The influence of three independent variables (duration, concentration, and ratio) at three levels (−1, 0, +1) and two different temperatures (4 °C and 21 °C) were examined (Table 4).

A total of 15 experiments with 3 replicas at the center point were performed.

To determine the acid conditions on collagen extraction the following parameters were tested: collagen yield (%), hydroxyproline yield (%) and SDS-PAGE pattern of collagen. Prior to acid extraction the skins were pre-treated with NaOH to remove non-collagen proteins. The pre-treated skins were removed and washed with tap water until it reached pH 7 and soaked in acetic acid with different extraction times, concentrations and ratios, as indicated in Table 4. After extraction the skin residues were filtered out with a sieve (pore size 100 µm) and the collagen in the solution was precipitated with NaCl at a final concentration of 2.6 M. The precipitate was collected by centrifugation and dialyzed three times using a dialysis bag with a molecular cut-off of 14 kDA (Sigma Aldrich Co., St. Louis MO, USA) and then freeze dried (Modulyo 4K Freeze Dryer, Ed-wards High Vacuum International, West Sussex, UK).

### 3.4. Verification of Found Optimal Alkali Pre-Treatment and Acid Extraction Parameters

The optimized conditions for the alkali pre-treatment and acid extraction were combined to investigate the effects of combined conditions. The thawed and shredded lumpfish skins (1 kg) were washed in 0.1 M NaOH at 4 °C for 6 h and a solid-to-liquid (*w*/*v*) ratio of 1:5. The residues were washed with tap water to reach pH 7 and acid extraction was performed with a solid-to-liquid (*w*/*v*) ratio 1:2 of 0.9 M acetic acid for 36 h at 21 °C. After extraction, the collagen in the suspension was collected, dialyzed and freeze-dried as described prior in Section 3.3. An additional purification step was performed on the freeze-dried product to obtain a higher quality. For that, the collagen was dissolved in ddH2O and dialyzed using a dialysis bag with a molecular cut-off of 10 kDA (Sigma Aldrich Co., St. Louis MO, USA) against tap water with change of water every 12 h. The pH of the tap water and the sample were monitored throughout the process. The dialysis was completed when the pH of the water was neutral. After dialysis, the sample was freeze-dried as described before. The collagen yield, hydroxyproline content, and SDS-PAGE patterns of all collagen samples were determined.

### 3.5. Upscaling to Pilot Scale

Following the established alkaline pre-treatment and acid extraction protocol a pilot scale extraction was performed on 45 kg lumpfish skin. The thawed skins were placed in a 1000 L stirring tank containing 0.1 M NaOH solution at 4 °C in a ratio 1:5 (*w*/*v*) for 6 h. Afterwards the skins were washed thoroughly until the draining water had pH of 5–7. The pre-treated skins were then put into a cool (4–10 °C) 0.9 M acetic acid solution (ratio 1:2 *w*/*v*) for 36 h. After acid extraction, the skins were divided into two parts: 34 kg was used for a second acid extraction and 10 kg for other experiments (e.g., enzyme hydrolysis (see Section 3.6). The 34 kg skin sample was placed in 0.9 M acetic acid (ratio 1:2 *w*/*v*) for 36 h and 21 °C under constant stirring.

The acetic acid solution obtained in the process was filtered through a tangential flow filtration (TFF) system using10 kD Pall low protein binding omega cassettes. The retentate and permeate were frozen. The moisture and hydroxyproline content were determined for the raw material, pre-treated skins, extracted skins and all solutions obtained.

### 3.6. Enzyme Hydrolysis

Enzyme hydrolysis was performed on pre-treated skins obtained in the pilot scale experiment as (I) an acid extraction with enzyme digestion and (II) an enzyme digestion only as shown in Figure 6. Both experiments were carried out in triplicates. For the acid extraction with enzyme digestion, 200 g of pre-treated lumpfish skin was extracted for 36 h at 21 °C stirring at 110 rpm in 0.9 M acetic acid (1:2 *w*/*v*). The obtained solution was treated as described previously in Section 3.4. For the following enzyme digestion, NaOH was added to the skin residues until pH 7. Then, the ESC was extracted in water containing 0.25% (of sample dry weight) of the enzyme Protamex^®^ (Tailorzyme, Herlev, Denmark), at a sample-to-liquid ratio of 70:30 (*w*/*v*) at 54 °C and 150 rpm for 120 min (Innova^®^ 42/ 42R Incubator Shaker, Eppendorf AG, 22331 Hamburg, Germany).

The suspension was centrifuged, and the precipitate was freeze-dried (Modulyo 4K Freeze Dryer, Ed-wards High Vacuum International, West Sussex, UK). For the enzyme digestion only, 200 g of pre-treated fish skins were treated as described above. NaOH was added until neutral pH, 0.25% enzyme was added, at a solid-to-liquid ratio of 70:30 (*w*/*v*) at 54 °C and 150 rpm for 120 min. (Incubator). The suspension was centrifuged, and the precipitate was freeze-dried (Modulyo 4K Freeze Dryer, Ed-wards High Vacuum International, West Sussex, UK). The hydroxyproline content of the freeze-dried samples was determined.

### 3.7. Analysis

Body and proximate composition (water, protein, lipid, and ash contents) of female lumpfish carcasses was determined. Removed protein and hydroxyproline content in the NaOH solutions after pre-treatment were determined. The hydroxyproline content in raw skins and extracted collagen was analyzed and SDS-PAGE was performed to gain knowledge about the molecular weight distribution of extracted collagen. The collagen yield was determined and is defined as the freeze-dried product after dialysis.

#### 3.7.1. Body and Proximate Composition

The average body composition was estimated from five female lumpfish captured in Iceland. The fish were manually separated into roe, head, backbone, skin, and meat (fillet). Proximate analyses were performed on the skin. The protein content was estimated by Kjeldahl method (conversion factor of N × 5.6) [31]. Fat was estimated using a Soxhlet method in Soxtec ST243 instrument (Foss, Hilleroed, Denmark). The dried sample was placed in pre-dried aluminum trays, hexane (40–50 mL) was added, and the sample was extracted for 20 min and then rinsed for 40 min. The cups were then dried for 30 min for 1 h at 104 °C and then weighed. Dry weight was estimated by drying the sample (5 g) in a pre-dried porcelain cup at 104 °C for 4 h [32]. Ash content was estimated by first drying a 10 g sample for 1 h and then placing the dried sample in an oven at 550 °C for 12 h–18 h before weighing [33].

#### 3.7.2. Protein Removal

The amount of protein removed from the lumpfish skins during pre-treatment with NaOH was determined in the NaOH solutions using Bradford method [34]. A standard curve with a final concentration of 125 µg/mL to 2000 µg/mL of BSA was prepared. Then, 5 µL of each standard and sample were added into microplate wells, and 250 µL of dye reagent was added. Samples were incubated at room temperature for 10 min; the absorbance was read at 595 nm using an Epoch Spectrophotometer (BioTek Instruments, Winoo-ski, VT, USA) and Gen5 V1.11.5 software to analyze the readings (BioTek Instruments, Winooski, VT, USA).

#### 3.7.3. Determination of Hydroxyproline Content

Fish skins were hydrolyzed directly, but solution samples were freeze dried prior to hydrolysis. Per 50 mg of protein, 1 mL of 6N HCl was used to hydrolyze each sample at 105 °C for 22 h. The hydroxyproline content in the hydrolysate was determined after the method of Leach [35] using L-hydroxyproline as standard (5–25 µg/mL). The absorbance was measured at 555 nm with an Epoch Spectrophotometer (BioTek Instruments, Winooski, VT, USA) and Gen5 V1.11.5 software to analyze the readings (BioTek Instruments, Winooski, VT, USA).

The hydroxyproline content (*HYP)* was calculated as follows:(1)$$HYP\;\left(\%\right)=\frac{h}{dw}\times 100$$
where:

*h* = hydroxyproline content in measured sample (mg);

*dw* = dry weight of measured sample (mg).

A conversion factor of 12.6 was used to determine the purity of collagen (*CP*) based on the hydroxyproline content:(2)$$CP\;\left(\%\right)=HYP\times 12.6$$

#### 3.7.4. SDS-PAGE

SDS-PAGE of collagen was performed in a Mini PROTEAN Tetra Sell system (Bio-Rad Laboratories, Inc., Hercules, CA, USA) [36], using 10% TGX Stain-FreeTM Fast CastTM acrylamide gels (Bio-Rad Laboratories, Inc., Hercules, USA). Samples of Section 3.3 and Section 3.4 were prepared by adding ddH2O to a final concentration of 5–10 µg/10 µL protein and adding an equal volume of 4 × SDS sample buffer (125 mM Tris-HCl, 2 mM EDTA, pH 6.8, 40% glycerol, 10% SDS, 4 mM DTT). 20 µL of each sample were loaded in per well. After electrophoresis, the gels were stained with 0.1% *w*/*v* Coomassie Brilliant Blue R 250 in 50% *v*/*v* methanol and 10% *v*/*v* glacial acetic acid for 30 min, and then destained in 5% *v*/*v* methanol and 7.5% *v*/*v* glacial acetic acid.

#### 3.7.5. Collagen Yields

The collagen yield (*CY*) was calculated as follows:(3)$$CY\;\left(\%\right)=\frac{SC}{DW}\times 100,$$
where:

*SC* = dry weight of extracted collagen (g);

*DW* = dry weight of raw lumpfish skins (g).

### 3.8. Statistical Analysis

All data were analyzed using R (4.3.1) and R Studio (2023.09.1) [37]. Values are given as mean values with standard deviations. Linear, quadratic and interaction models were tested to find the best fitting model. Analysis of variance (ANOVA) was used to determine the significance of each model (*p* < 0.05).

## 4. Conclusions

It was initially hypothesized that the collagen extraction yield from lumpfish skin could surpass the previously reported range of 14–40%. The optimized protocol developed in this study achieved yields up to 60%, indicating significant improvement. Results suggest that the comprehensive optimization of each stage—from grinding and pre-cleaning to acid extraction and purification—has successfully enhanced extraction yield, purity, and hydroxyproline content. This was achieved by increasing the acid extraction time (36 h) and using 0.9 M acetic acid and adding an enzyme extraction step. These findings represent the highest reported yields to date, providing a strong foundation for future scale-up investigations.

## Figures and Tables

**Figure 1 marinedrugs-22-00525-f001:**
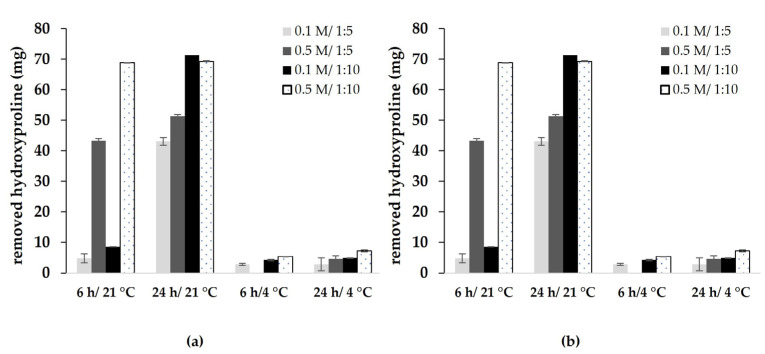
Total amount (g) of proteins removed (**a**) and amount (mg) of hydroxyproline removed (**b**) by NaOH pre-treatment of 30 g skin samples at different conditions.

**Figure 2 marinedrugs-22-00525-f002:**
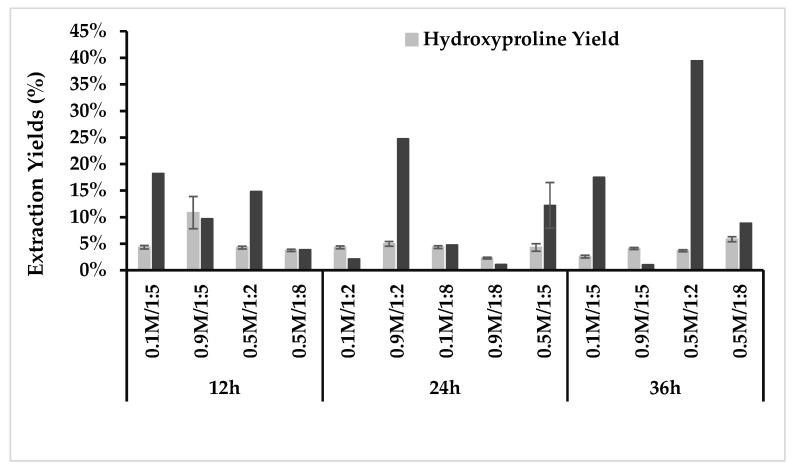
Collagen and hydroxyproline extraction yields (%) (DW/DW) of samples extracted at 21 °C with acetic acid at different concentrations, ratios (*w*/*v*) and extraction times. Hydroxyproline yield is shown as mean ± SD. The center point at 0.5 M/1:5/24 h of the Box–Behnken design was performed in triplicate and is shown as an average with a standard deviation of 4% for the collagen yield.

**Figure 3 marinedrugs-22-00525-f003:**
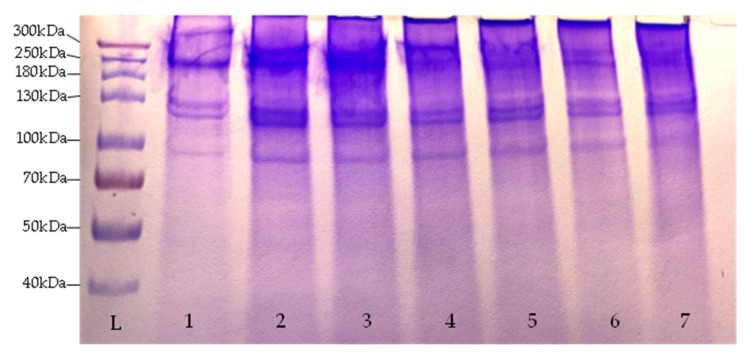
SDS-PAGE patterns of lumpfish skin collagen extracted at 21 °C. Lane L: molecular weight marker, lane 1–7: samples extracted for 24 h.

**Figure 4 marinedrugs-22-00525-f004:**
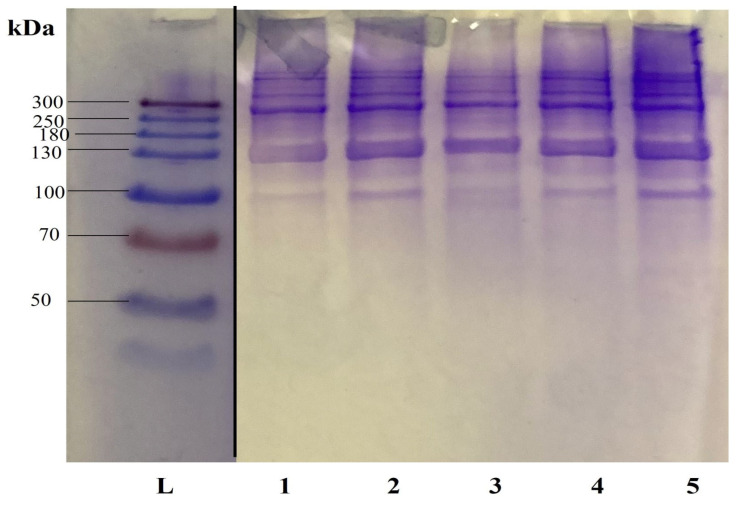
SDS-page patterns of lumpfish skin collagen from 1 kg extraction. Lane L: molecular weight marker, lane 1–5: samples extracted for 36 h. Gel sliced at black line.

**Figure 5 marinedrugs-22-00525-f005:**
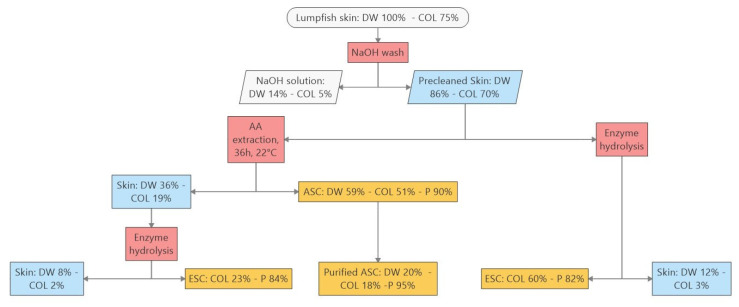
Mass flow of dry weight (DW g) and collagen (COL, %) during acetic acid (AA) extraction and enzyme digestion of lumpfish skin. 100% DW equals 15.4 g at start and COL % is the collagen DW/DW yield. Collagen purity (P) is calculated as HYP% × 12.6. Acid Soluble Collagen (ASC); Enzyme Soluble Collagen (ESC).

**Figure 6 marinedrugs-22-00525-f006:**
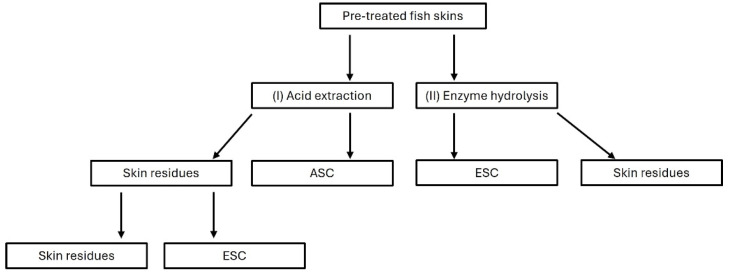
Workflow for the extraction of Acid Soluble Collagen (ASC) and Enzyme Soluble Collagen (ESC) from pre-treated lumpfish skins.

**Table 1 marinedrugs-22-00525-t001:** Extraction yields of the 5 fractions obtained after salting out the collagen extracted from 1 kg of lumpfish skin (110.03 g DW/DW). ^1^ based on the conversion factor of 12.6.

Fraction	Dry Weight (g)	Dry Weight Yield (DW%)	Hydroxyproline (%)	Collagen Yield ^1^ (DW%)	Collagen Purity (%)
1	0.33	0.30	6.28 ± 0.55	0.24 ± 0.02	79.13 ± 6.93
2	1.71	1.55	6.78 ± 0.50	1.33 ± 0.10	85.43 ± 6.30
3	0.03	0.03	6.53 ± 0.59	0.02 ± 0.00	82.28 ± 7.43
4	2.02	1.83	7.20 ± 0.69	1.67 ± 0.16	90.72 ± 8.69
5	49.20	44.71	6.69 ± 0.61	37.70 ± 3.44	84.29 ± 7.69
5 purified			7.92 ± 0.80		99.79 ± 10.08

**Table 2 marinedrugs-22-00525-t002:** Yield of hydroxyproline (HYP), collagen (COL) and mass balance from acetic acid extraction of collagen from lumpfish skins (*n* = 3). Wet weight (WW), dry weight (DW) and no data (n.d.).

Sample	WW (g)	DW (%)	DW (g)	HYP (%) ± SD	COL (g) ± SD	Purity (%) ± SD	Collagen Yield (%) (DW/DW)
Skin at start	45,000	14.0	6435	5.1 ± 1.8	4127 ± 1390	64 ± 22	
Skin before wash	34,000	14.3	4862	5.1 ± 1.8	3118 ± 1050	64 ± 22	64
Skin after NaOH wash	50,000	8.4	4200	5.5 ± n.d.	2911 ± n.d.	69 ± n.d.	60
Skin after AcOH extraction	26,900	5.4	1442	6.3 ± 0.5	1145 ± 87	79 ± 6	24
Extraction liquid before filtration	119,400	1.3	1493	6.7 ± 0.6	1258 ± 107	84 ± 7	26
Permeate	87,900	0.1	79	0.0 ± 0	0.0 ± 0	0.0 ± 0	0
Retentate	17,800	6.6	1173	5.9 ± 0.5	872 ± 70	74 ± 6	18

**Table 3 marinedrugs-22-00525-t003:** Experimental range and values of independent variables for pre-treatment of lumpfish skins.

Factor	Coded Values	
	−1	+1
	**Real values**	
t (h)	6	24
NaOH (M)	0.1	0.5
T (°C)	4	21
ratio (*w*/*v*)	1:5	1:10

**Table 4 marinedrugs-22-00525-t004:** Experimental range and values of independent variables in the Box–Behnken design with coded values (−1, 0, and +1) and corresponding real values for the acetic acid extraction of lumpfish skins.

Factor	Coded Values		
	−1	0	+1
	**Real values**		
duration (h)	12	24	36
AcOH (M)	0.1	0.5	0.9
ratio (*w*/*v*)	1:2	1:5	1:8

## Data Availability

Data is available from the corresponding author.

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
