# Peer review of "Maximizing Collagen Yield from Underutilized Lumpfish (Cyclopterus lumpus) Skins by Optimizing Pre-Cleaning and Extraction Methods"

_marinedrugs, 2024, doi:10.3390/md22120525_

Round 1
Reviewer 1 Report
Comments and Suggestions for Authors
In this study, Scheja et al. aimed to maximize the collagen yield extracted from lumpfish skin and enhance the value of byproducts from the lumpfish fishing industry. The authors optimized pre-treatment and acid extraction conditions for collagen extraction from lumpfish skin. The study was well conducted, especially the scaled-up experimental study, which brought this study closer to the actual production applications. However, there are some points that need further clarification. Here are some comments on this study:
1. In the introduction section, it is recommended that the authors could state what the gaps or shortcomings in the current research that motivated the authors to conduct the present study.
2. There are some inappropriate placements of references in some sentences, such as line 65, line 69, and line 383 “[29]”.
3. Line 79 “ANOVA”, did the authors test the distribution and variance of the data?
4. It is recommended that the authors label the figures with the results of the statistical analysis of differences.
5. Line 111 “Sato et al. [18] [18]”.
6. Lines 219-220, it is suggested that the authors combine this sentence with the next paragraph.
7. It is recommended that the authors could rewrite the conclusion section, currently, some sentences seem confusing, especially “The results (e.g. yield and purity) are important information regarding further eval-394 uation of the feasibility of full utilization of the vast amount of lumpfish by-products.”.
Author Response
We appreciate the Editor and Reviewers for their valuable time in assessing our work and for their constructive feedback, which has been instrumental in enhancing the manuscript. We have carefully addressed each point raised to improve the quality of our submission.
|
Comment 1 In the introduction section, it is recommended that the authors could state what the gaps or shortcomings in the current research that motivated the authors to conduct the present study. |
|
Response 1 Two paragraphs have been added to the introduction to address this comment. First, the inconsistency (variation/range) in results regarding collagen yield from different species especially lumpfish is highlighted. Secondly the diversity and importance of extraction protocols on yield results are also highlighted. In our opinion this highly motivate (at least us) to look more closely into these factors. The paragraph on aim (L 72-74) has been modified to underpin the work better. We would also like to point out that there are only one or two reports on collagen extraction from lumpfish that are bigger than a lab scale experiment, so that is new. |
|
Comment 2 There are some inappropriate placements of references in some sentences, such as line 65, line 69, and line 383 “[29]”. |
|
Response 2 This has been reviewed and changed accordingly. |
|
|
|
Comment 3 Line 79 “ANOVA”, did the authors test the distribution and variance of the data? |
|
Response 4 We did the Shapiro-Wilk normality test which confirmed normal distribution of ANOVA and Barlett test confirmed homogeneity of variances. |
|
|
|
Comment 4 It is recommended that the authors label the figures with the results of the statistical analysis of differences. |
|
Response 4 That is a good point, and we tried to come up with a good way to label the figure with the results of the ANOVA. However, because it’s a four-way ANOVA (i.e. four factors) we found that the labelling becomes too confusing. |
|
|
|
Comment 5 Line 111 “Sato et al. [18] [18]”. |
|
Response 5 This has been changed accordingly. |
|
|
|
Comment 6 Lines 219-220, it is suggested that the authors combine this sentence with the next paragraph. |
|
Response 6 Changes have been made. |
|
|
|
Comment 7 It is recommended that the authors could rewrite the conclusion section, currently, some sentences seem confusing, especially “The results (e.g. yield and purity) are important information regarding further eval-394 uation of the feasibility of full utilization of the vast amount of lumpfish by-products.”. |
|
Comment 7 The Conclusions have been rewritten to address this comment. Now the Conclusions have a “reference” to the hypothesis set in the introduction, i.e. we conclude that we could improve the yield. The sentence pointed out by the reviewer has been deleted. As we are considering further development of this project toward bigger pilot scale collagen recovery from lumpfish the yield, we obtained here are very encouraging. |
Reviewer 2 Report
Comments and Suggestions for Authors
In the manuscript (ID: marinedrugs-3296651), authors aimed to research the production, purification and antioxidant activities of Bacteria-derived natural metabolites. Generally, the content of this manuscript meets the requirements of Marine Drugs. Therefore, I think this manuscript is suitable for publication in the journal of Marine Drugs after a major revision.
(1) In general, this manuscript is a common study. The methods used in this manuscript are quite popular. Therefore, the author needs to explain the significance and innovation of this study in Introduction. In addition, in the results, the author needs to explain the differences between the results of this study and the results of other studies, and more importantly, the results of this study exceed the expectations of other studies..
(2) Page 1, line 1: Should be “Article” rather “Article,”.
(3) Page 1, Title: Should be “skins by optimizing pre-cleaning and extraction methods” rather skin by optimizing pre-cleaning and extraction methods.”.
(4) Page 1, line 13: Should be “0.1 M (NaOH concentration), X2 = 6 h (NaOH treatment time), X3 = 4 °C” rather “0.1M (NaOH concentration), X2 = 6h (NaOH treatment time), X3 = 4°C”.
(5) Page 1, line 16: Should be “0.9 M acetic acid, treatment temperature of 21 °C, a treatment time of 36 h” rather “0.9M acetic acid, treatment temperature of 21°C, a treatment time of 36h”. In addition, there are similar errors in other parts of the manuscript, and authors are advised to check the whole manuscript carefully and correct these minor errors.
(6) In this Introduction, the review on the high-value utilization of aquatic by-products, especially the extraction of collagen, gelatin and collagen peptides from aquatic by-products, lacks breadth and depth. Presently, some literatures have conducted more detailed studies on the utilization of aquatic by-products to produce collagen, gelatin and collagen peptides, such as acid-soluble collagen from the skin of hammerhead shark (Sphyrna lewini), collagens from the scales of miiuy croaker (Miichthys miiuy), collagens from the cartilage of Siberian sturgeon, collagens from the swim bladders of miiuy croaker (Miichthys miiuy), collagens from scales of croceine and redlip croakers, gelatins and antioxidant peptides from Skipjack tuna (Katsuwonus pelamis) skins, Gelatin and antioxidant peptides from gelatin hydrolysate of skipjack tuna (Katsuwonus pelamis) scales, etc. It is suggested that the authors systematically review the activity and structure of these previous researches, so as to further explain the innovation, importance and significance of this work.
(7) Page 1, line 27: Should be “where of” rather “whereof”.
(8) Page 1, line 37: Should be “[6,7]” rather “[6][7]”.
(9) Page 2, line 47-49: Each polypeptide chain is made up of a repeated sequence (Gly-X-Y-), with most commonly proline in X and 4-hydroxyproline in Y-position. Hydroxyproline is found in collagen and in small amounts in elastin but is nearly absent in all other proteins. Authors are suggested to add literatures to support the content.
(10) Page 2, line 52-56: Here, the author only explained the extraction steps of collagen, which completely failed to meet the purpose of summarizing the existing studies. It is suggested that the author can make an in-depth analysis of the existing methods here, so as to explain their advantages and disadvantages, and further explain the significance and innovation of this study.
(11) Page 2, line 63: Should be “In comparison to the study of Sato et al.” rather “In comparison Sato et al.”.
(12) Page 2, line 69: Should be “[8,16]” rather “[8] [16]”. In addition, there are similar errors in other parts of the manuscript, and authors are advised to check the whole manuscript carefully and correct these minor errors.
(13) Page 3, Figure 1: Data analysis, especially significance analysis, should be applied to all experimental results of the manuscript.
(14) Page 3, line 100: Should be “The factors temperature (21 °C), time (24 h) and concentration (0.5 M)” rather “The factors temperature (21°C), time (24h) and concentration (0.5M)”.
(15) Page 3, line 105: p < 0.05. “p” should be in italics. In addition, there are similar errors in other parts of the manuscript, and authors are advised to check the whole manuscript carefully and correct these minor errors.
(16) Page 3, line 111: by Sato et al. [18] [18]? Please check this information carefully.
(17) Page 4, Figure 2: The data should be expressed as mean ± SD. In addition, Data analysis, especially significance analysis, should be applied to all experimental results of the manuscript.
(18) Page 5, Figure 3: If possible, this SDS-PAGE should include a sample of standard Type I collagen. Otherwise, how can the author judge that the extracted collagen belongs to type I collagen?
Comments on the Quality of English Language
The language needs to be improved both grammatically and scientifically.
Author Response
|
We appreciate the Editor and Reviewers for their valuable time in assessing our work and for their constructive feedback, which has been instrumental in enhancing the manuscript. We have carefully addressed each point raised to improve the quality of our submission. |
|
Comment 1: In general, this manuscript is a common study. The methods used in this manuscript are quite popular. Therefore, the author needs to explain the significance and innovation of this study in Introduction. In addition, in the results, the author needs to explain the differences between the results of this study and the results of other studies, and more importantly, the results of this study exceed the expectations of other studies. |
|
Response:1 Two paragraphs have been added to the introduction to address this comment. First, the inconsistency (variation/range) in results regarding collagen yield from different species especially lumpfish is highlighted. Secondly the diversity and importance of extraction protocols on yield results are also highlighted. In our opinion this highly motivate (at least us) to look more closely into these factors. The paragraph on aim (L 72-74) has been modified to underpin the work better. We would also like to point out that there are only one or two reports on collagen extraction from lumpfish that are bigger than a lab scale experiment, so that is new. |
|
Comment 2 Page 1, line 1: Should be “Article” rather “Article,”. |
|
Response 2 This has been changed accordingly. |
|
Comment 3 Page 1, Title: Should be “skins by optimizing pre-cleaning and extraction methods” rather skin by optimizing pre-cleaning and extraction methods.”. |
|
Response 3 Change has been made. |
|
Comment 4 Page 1, line 13: Should be “0.1 M (NaOH concentration), X2 = 6 h (NaOH treatment time), X3 = 4 °C” rather “0.1M (NaOH concentration), X2 = 6h (NaOH treatment time), X3 = 4°C”. |
|
Response 4 This has been changed accordingly. |
|
Comment 5 Page 1, line 16: Should be “0.9 M acetic acid, treatment temperature of 21 °C, a treatment time of 36 h” rather “0.9M acetic acid, treatment temperature of 21°C, a treatment time of 36h”. In addition, there are similar errors in other parts of the manuscript, and authors are advised to check the whole manuscript carefully and correct these minor errors |
|
Response 5 Thank you for pointing these out, we checked and changed accordingly. |
|
Comment 6 In this Introduction, the review on the high-value utilization of aquatic by-products, especially the extraction of collagen, gelatin and collagen peptides from aquatic by-products, lacks breadth and depth. Presently, some literatures have conducted more detailed studies on the utilization of aquatic by-products to produce collagen, gelatin and collagen peptides, such as acid-soluble collagen from the skin of hammerhead shark (Sphyrna lewini), collagens from the scales of miiuy croaker (Miichthys miiuy), collagens from the cartilage of Siberian sturgeon, collagens from the swim bladders of miiuy croaker (Miichthys miiuy), collagens from scales of croceine and redlip croakers, gelatins and antioxidant peptides from Skipjack tuna (Katsuwonus pelamis) skins, Gelatin and antioxidant peptides from gelatin hydrolysate of skipjack tuna (Katsuwonus pelamis) scales, etc. It is suggested that the authors systematically review the activity and structure of these previous researches, so as to further explain the innovation, importance and significance of this work. |
|
Response 6 Here we would again point out the changes made in the Introduction (under point 1). We have added new references (Farooq et al. and Laasri et al that verry well describe collagen yield from different species and tissues. We have also included in the introduction references to lumpfish collagen, those were (and still are) in the Result and discussion part. Also, we have changed the text in conclusions to try to address this comment. |
|
Comment 7 Page 1, line 27: Should be “where of” rather “whereof”. |
|
Response 7 Modified accordingly. |
|
Comment 8 Page 1, line 37: Should be “[6,7]” rather “[6][7]”. |
|
Response 8 Modified accordingly. |
|
Comment 9 Page 2, line 47-49: Each polypeptide chain is made up of a repeated sequence (Gly-X-Y-), with most commonly proline in X and 4-hydroxyproline in Y-position. Hydroxyproline is found in collagen and in small amounts in elastin but is nearly absent in all other proteins. Authors are suggested to add literatures to support the content. |
|
Response 9 Literature was added. |
|
Comment 10 Page 2, line 52-56: Here, the author only explained the extraction steps of collagen, which completely failed to meet the purpose of summarizing the existing studies. It is suggested that the author can make an in-depth analysis of the existing methods here, so as to explain their advantages and disadvantages, and further explain the significance and innovation of this study. |
|
Response 10 Here again we would like to point out the changes already mentioned under point 1 and 6. Also the changes made to the conclusions. |
|
Comment 11 Page 2, line 63: Should be “In comparison to the study of Sato et al.” rather “In comparison Sato et al.”. |
|
Response 11 We have changed the sentence. |
|
Comment 12 Page 2, line 69: Should be “[8,16]” rather “[8] [16]”. In addition, there are similar errors in other parts of the manuscript, and authors are advised to check the whole manuscript carefully and correct these minor errors. |
|
Response 12 We added the changes throughout the document. |
|
Comment 13 Page 3, Figure 1: Data analysis, especially significance analysis, should be applied to all experimental results of the manuscript. |
|
Response 13 We did the Shapiro-Wilk normality test which confirmed normal distribution of ANOVA and Barlett test confirmed homogeneity of variances |
|
Comment 14 Page 3, line 100: Should be “The factors temperature (21 °C), time (24 h) and concentration (0.5 M)” rather “The factors temperature (21°C), time (24h) and concentration (0.5M)”. |
|
Response 14 Modified accordingly. |
|
|
|
Comment 15 Page 3, line 105: p < 0.05. “p” should be in italics. In addition, there are similar errors in other parts of the manuscript, and authors are advised to check the whole manuscript carefully and correct these minor errors. |
|
Response 15 Modified accordingly. |
|
|
|
Comment 16 Page 3, line 111: by Sato et al. [18] [18]? Please check this information carefully. |
|
Response 16 We checked and corrected accordingly. |
|
|
|
Comment 17 Page 4, Figure 2: The data should be expressed as mean ± SD. In addition, Data analysis, especially significance analysis, should be applied to all experimental results of the manuscript. |
|
Response 17 Thank you for pointing this out. We added the means ±SD to the results. Wew did add the SD to the OHProline yield, but not to the collagen yield as this is the result of the weight of the freezed- dried material and the DW of the starting material, the Box Behnken Design does not require duplicates so no SD here. |
|
|
|
Comment 18 Page 5, Figure 3: If possible, this SDS-PAGE should include a sample of standard Type I collagen. Otherwise, how can the author judge that the extracted collagen belongs to type I collagen? |
|
Response 18 Unfortunately, this is not possible. But an extensive literature review let us believe that this collagen might be type I collagen. The molecular weight pattern of the bands is similar to literature of other fish species as well as lumpfish. We added additional sources as well as text line 165pp. |
Round 2
Reviewer 2 Report
Comments and Suggestions for Authors
The authors have made commendable revisions to the manuscript (ID marinedrugs-3296651), addressing several issues effectively. However, the issue that still need to be concerned before the manuscript can be accepted for publication in Marine Drugs.
(1) Figure 3 and Figure 4: The SDS-PAGE should include a sample of standard Type I collagen. Otherwise, it cannot be concluded that the prepared collagen is type I collagen. In addition, the authors need more methods to prove that the prepared collagen is type I collagen.
Comments on the Quality of English Language
The English could be improved to more clearly express the research.
Author Response
Comment 1: Figure 3 and Figure 4: The SDS-PAGE should include a sample of standard Type I collagen. Otherwise, it cannot be concluded that the prepared collagen is type I collagen. In addition, the authors need more methods to prove that the prepared collagen is type I collagen.
Reply 1: The authors acknowledge that in the absence of collagen standard it can not be concluded that the extracted collagen is of type I. Our conclusion was based on information from literature saying that collagen types in skin are mainly type I or type III. Comparing our results with those in literature lead to our conclusion that we had extracted type I. We have changed sentences and figure texts were we mention collagen type I and no longer claim that we have extracted type I but still we indicates that this might be so.
See lines: 18-19, 165-166, 169, 196,
Comment 2: The English could be improved to more clearly express the research.
Reply 2: The manuscript will undergo copy editing
Round 3
Reviewer 2 Report
Comments and Suggestions for Authors
The authors have carefully revised the manuscript (ID: marinedrugs-3296651) and the quality of the manuscript has been improved accordingly. Therefore, I think that the manuscript can be accepted for publication in Marine Drugs.